# The Roles of MicroRNAs in Asthma and Emerging Insights into the Effects of Vitamin D_3_ Supplementation

**DOI:** 10.3390/nu16030341

**Published:** 2024-01-24

**Authors:** Adrián Hernández-Díazcouder, Rodrigo Romero-Nava, Blanca E. Del-Río-Navarro, Fausto Sánchez-Muñoz, Carlos A. Guzmán-Martín, Nayely Reyes-Noriega, Octavio Rodríguez-Cortés, José J. Leija-Martínez, Juan Manuel Vélez-Reséndiz, Santiago Villafaña, Enrique Hong, Fengyang Huang

**Affiliations:** 1Laboratorio de Investigación de Obesidad y Asma, Hospital Infantil de México Federico Gómez, Ciudad de Mexico 06720, Mexico; adrian.hernandez.diazc@hotmail.com (A.H.-D.); nreyes@himfg.edu.mx (N.R.-N.); 2Instituto Mexicano del Seguro Social, Hospital de Especialidades “Dr. Bernardo Sepúlveda Gutiérrez”, Unidad de Investigación Médica en Bioquímica, Ciudad de Mexico 06720, Mexico; 3Laboratorio de Señalización Intracelular, Sección de Estudios de Posgrado e Investigación, Escuela Superior de Medicina, Instituto Politécnico Nacional, Ciudad de Mexico 11340, Mexico; rromeron@ipn.mx (R.R.-N.); svillafana@ipn.mx (S.V.); 4Servicio de Alergia e Inmunología, Hospital Infantil de México Federico Gómez, Ciudad de Mexico 06720, Mexico; berio@himfg.edu.mx; 5Departamento de Inmunología, Instituto Nacional de Cardiología Ignacio Chávez, Ciudad de Mexico 14080, Mexico; fausto.sanchez@cardiologia.org.mx (F.S.-M.); 2231801024@alumnos.xoc.uam.mx (C.A.G.-M.); 6Laboratorio de Inflamación y Obesidad, Sección de Estudios de Posgrado e Investigación, Escuela Superior de Medicina, Instituto Politécnico Nacional, Ciudad de Mexico 11340, Mexico; orodriguezc@ipn.mx; 7Centro de Investigación en Ciencias de la Salud y Biomedicina, Universidad Autónoma de San Luis Potosí, San Luis Potosí 78290, Mexico; jesus.leija@uaslp.mx; 8Laboratorio Multidisciplinario de Nanomedicina y de Farmacología Cardiovascular, Sección de Estudios de Posgrado e Investigación, Escuela Superior de Medicina, Instituto Politécnico Nacional, Ciudad de Mexico 11340, Mexico; jvelezr@ipn.mx; 9Departamento de Farmacobiología, Centro de Investigación y Estudios Avanzados del Instituto Politécnico Nacional, Ciudad de Mexico 14330, Mexico; ehong@cinvestav.mx

**Keywords:** asthma, inflammation, microRNAs, vitamin D_3_ supplementation

## Abstract

Asthma is one of the most common chronic non-communicable diseases worldwide, characterized by variable airflow limitation secondary to airway narrowing, airway wall thickening, and increased mucus resulting from chronic inflammation and airway remodeling. Current epidemiological studies reported that hypovitaminosis D is frequent in patients with asthma and is associated with worsening the disease and that supplementation with vitamin D_3_ improves asthma symptoms. However, despite several advances in the field, the molecular mechanisms of asthma have yet to be comprehensively understood. MicroRNAs play an important role in controlling several biological processes and their deregulation is implicated in diverse diseases, including asthma. Evidence supports that the dysregulation of miR-21, miR-27b, miR-145, miR-146a, and miR-155 leads to disbalance of Th1/Th2 cells, inflammation, and airway remodeling, resulting in exacerbation of asthma. This review addresses how these molecular mechanisms explain the development of asthma and its exacerbation and how vitamin D_3_ may modulate these microRNAs to improve asthma symptoms.

## 1. Introduction

Asthma is one of the most common chronic non-communicable diseases worldwide, affecting 1–18% of the populations of different countries [1]. In 2017, asthma incidence worldwide was 43.13 million new cases/year, while the prevalence was 272.68 million cases and a mortality of 0.49 million deaths [2]. Moreover, asthma prevalence is higher in industrialized countries than in low-income and middle-income countries [1] For instance, in the United States, asthma prevalence increased from 20 million (7.3%) to 25 million (8.0%) from 2001 to 2017 [3]. Likewise, asthma prevalence was slightly higher in US children (8.4%) than in US adults (7.7%) [3]. In LATAM, findings of the International Study of Asthma and Allergies in Childhood reported an asthma prevalence of 18% among children aged 13 to 14 years [4]. While higher-income nations tend to have greater asthma prevalence rates, it is noteworthy that the majority of asthma-related deaths occur in lower to middle-income regions like LATAM, where the asthma mortality rate stands at 26.3 per 100,000 individuals [4].

Asthma is a common chronic airway disease characterized by variable airflow limitation secondary to airway narrowing, airway wall thickening, and increased mucus [5], resulting from chronic inflammation and airway remodeling [5,6]. These alterations lead to airway hyperresponsiveness (AHR), airway obstruction, airflow limitation, and progressive decline of patients’ lung function [7]. Asthma is a heterogeneous disease with several distinct clinical presentations (phenotypes) and complex pathophysiological mechanisms (endotypes) [6]. Based on endotypes, asthma can be categorized into diverse classifications, including non-allergic and allergic, non-eosinophilic and eosinophilic, type 2 (T2) high and T2 low, or its equivalent non-T2, with respect to the inflammatory profile [6]. T2 inflammation involves the innate (type 2 innate lymphoid cell (ILC2)) and adaptative (T-helper type 2 cells (Th2)) immune system [8]. When ILC2 and Th2 cells are triggered by contact with an allergen, they produce type-2 cytokines (interleukin (IL)-4, IL-5, and IL-13). IL-4 and IL-13 induce B cell class switching and IgE production, the release of pro-inflammatory mediators, barrier disruption, and tissue remodeling; IL-13 induces goblet-cell hyperplasia and mucus production [9]. Altogether, these cytokines recruit eosinophils to tissues, generating clinical symptoms of chronic inflammatory airway diseases [9]. This complex pathophysiology coupled with the global burden of asthma underscores the urgent need for the exploration of novel therapeutic strategies. Among these, the potential role of vitamin D_3_ in the management of asthma presents a promising avenue for exploration.

Vitamin D_3_ [1,25-dihydroxy vitamin D (1,25(OH)2D), also called calcitriol] may offer therapeutic benefits in asthma through multifaceted mechanisms. Its anti-inflammatory properties encompass the modulation of immune responses, particularly the regulation of pro-inflammatory cytokines linked to asthma through vitamin D receptor (VDR) signaling [10,11]. Recent epidemiological studies have reported a relationship between vitamin D deficiency/insufficiency and asthma [12,13,14,15] and that supplementation with vitamin D_3_ has beneficial effects on the development and exacerbation of asthma [16,17]. Despite several advances in the field, the molecular mechanisms related to the beneficial effects of vitamin D in asthma are minimal. Notably, vitamin D_3_’s impact on microRNA (miRNA) expression highlights its role in regulating known molecular key pathways associated with asthma.

In this sense, miRNAs represent a plausible molecular mechanism through which vitamin D_3_ exerts its beneficial effects on the inflammatory response. These small non-coding RNAs are pivotal in post-transcriptional gene regulation [18] and have been increasingly recognized for their role in modulating immune responses [19]. Notably, vitamin D_3_ supplementation has been observed to alter the miRNA expression profile in both plasma and cells [20,21]. This modulation of miRNA expression is particularly relevant in the context of diseases like asthma, where miRNA deregulation is a common feature.

Emerging evidence underscores that miRNAs are differentially expressed in individuals with asthma compared to those without, highlighting their significant immunoregulatory roles [22]. These miRNAs are not only cell-/tissue-specific but have also been directly associated with asthmatic pathology [23,24,25]. Within T helper (Th) cell subpopulations, including Th1, Th2, Th17, and Treg cells, distinct miRNA signatures have been identified [26]. These signatures not only reflect the activation status of these cells but also the nature of inflammation characterizing various asthma phenotypes/endotypes and their severity levels. Targeting these miRNAs, especially through the strategic use of vitamin D_3_ supplementation, could offer a novel approach to modulate the inflammatory state in asthma.

In this review, we will address epidemiological studies of the relationship between vitamin D deficiency/insufficiency and asthma and the molecular mechanisms mediated by miRNAs, which are implicated in disbalance Th1/Th2 cells, inflammatory response, and airway remodeling, as well as its possible regulation by supplementation with vitamin D_3_ as a therapeutic strategy.

## 2. The Relationship between Vitamin D Insufficiency and Asthma

According to the Society for Endocrinology, vitamin D sufficiency is defined as when serum 25(OH)D concentrations are ≥30 ng/mL. In contrast, insufficiency and deficiency concentrations are 20–29 ng/mL and <20 ng/mL, respectively [27]. Current epidemiological studies reported a strong relationship between hypovitaminosis D and asthma. For instance, Zhu Yiqun and collaborators reported that asthmatic and non-asthmatic subjects with insufficient and optimal vitamin D concentrations presented a decrease in odds of asthma by 6.4% and 9.8%, respectively, in comparison to subjects with vitamin D deficiency [12]. Likewise, they reported that each nmol/L increase in vitamin D in asthmatic patients was positively associated with a 1–2 mL increase in forced expiratory volume (FEV) or forced vital capacity (FVC) [12]. Moreover, asthmatic patients with vitamin D deficiency presented 19.9% higher odds of current wheezing than patients with insufficient and optimal vitamin D concentration [12]. A study in children conducted by Doumat George and collaborators reported an association between serum 25(OH)D concentration at age three and spirometry at age six. The children in the lowest quintile (8.4–20.1 ng/mL of 25(OH)D) presented a 6% and 7% lowest in FEV_1pp_ and FVC_pp_, respectively, and those in the fourth quintile (27.8–31.8 ng/mL of 25(OH)D) presented a 5% lower FEV_1pp_ in comparison with children in the highest quintile (31.5–53.0 ng/mL of 25(OH)D) [13]. Bhat K.G. and collaborators reported that children with intermittent asthma presented a mean of 28.52 ng/mL of vitamin D as compared to lower levels of 14.4 ng/mL and 12.5 ng/mL in the mild intermittent and moderate asthmatic children, respectively [14]. Moreover, those children with vitamin D deficiency/insufficient presented higher rates of exacerbations and hospitalization and increased use of rescue medications compared to those asthmatic children with vitamin sufficient [14]. Malheiro A.P. and collaborators found that children with severe asthma presented low concentrations of vitamin D (mean 24.28 ng/mL) in comparison to children with mild or moderate asthma (mean 27.04 ng/mL) [15]. Also, they found a positive correlation between vitamin D concentrations and lung function parameters (FEV_1_ and forced expiratory flow (FEF) 25–75%) [15]. Likewise, a meta-analysis that includes 35 studies showed that asthmatic children presented lower 25-OHD levels (21.7 ng/mL) than healthy children (26.5 ng/mL) [16]. The same study reported that supplementation with vitamin D_3_ reduced the recurrence rate in asthmatic children compared to asthmatic children with placebo [16]. Another meta-analysis that includes 14 studies reported that the supplementation with vitamin D in asthmatic patients with vitamin D insufficiency had a beneficial effect in improving FEV_1%_, airflow limitation, and asthma exacerbations [17]. The above evidence suggests that hypovitaminosis D could be an important risk factor for the development of asthma and its exacerbation. Furthermore, supplementation with vitamin D_3_ may be a therapeutic strategy for asthma due to its beneficial effects on pulmonary function. However, the molecular mechanisms underlying the beneficial effects of vitamin D in asthma, particularly those associated with miRNAs, are not well understood.

## 3. The Role of MicroRNA-21 in Asthma and Its Modulation by Vitamin D_3_

miR-21 is an important miRNA regulating the inflammatory response in various diseases, including asthma. In clinical studies, for instance, a recent study conducted on asthmatic patients reported increased levels of miR-21-5p in exosomes in patients with moderate asthma compared to those with healthy subjects, while patients with severe asthma presented reduced levels of miR-21-5p in exosomes [28]. Likewise, a recent study on patients with mild and moderate–severe asthma reported that moderate–severe asthma patients presented elevated levels of miR-21-5p in exosomes from the plasma compared with both mild asthma and healthy subjects [29]. Also, the same study reported that the levels of miR-21-5p in exosomes correlated positively with IgE levels and correlated negatively with plasma levels of TNF-α and IL-6 in patients with moderate–severe asthma [29]. Another recent study on asthmatic patients reported that asthmatic patients presented high levels of miR-21-5p and IL-4 in serum compared to healthy subjects [30]. The same study found a positive correlation between miR-21-5p levels and IL-4 levels; likewise, IL-4 predicted miR-21-5p expression levels [30]. A study with children reported that asthmatic children presented an increase in 42.6-fold in miR-21-5p plasma levels and an increase in IL-13 levels compared to healthy children [31]. Also, the same study reported that the plasma levels of miR-21-5p correlated positively with IL-13 levels and eosinophil percentage and correlated negatively with lung function (FEV1) [31]. In contrast, a study conducted on patients with a crossover phenotype of bronchial asthma (AB) and chronic obstructive pulmonary disease (COPD) reported that in those patients, there were decreased circulating levels of miR-21-5p compared with healthy subjects [32]. Furthermore, the same study found that low levels of miR-21-5p were associated with less reversibility of bronchial obstruction [32]. Therefore, evidence suggests that miR-21-5p participates in allergic lung inflammation and the pathogenesis of asthma.

In experimental studies within animal models, miR-21-5p was implicated in the development and exacerbation of asthma. For instance, in a recent study with a rat asthma model induced by ovalbumin (OVA), the expression of miR-21-5p was observed mainly in alveolar macrophages [33]. To validate the above result, rat alveolar macrophages were stimulated with LPS (to mimic an inflammatory state present in asthma), resulting in the increased expression of miR-21-5p in exosomes [33]. Besides, the transfer of the miR-21-5p-enriched exosomes from alveolar macrophages to rat airway epithelial cells induced an epithelial–mesenchymal transition through the regulation of Smad7 (Figure 1a) [33]. Moreover, it was reported that the expression of miR-21-5p in the human airway smooth muscle (ASM) cells induced an increase in the appearance and migration of these cells because of the regulation of homologous phosphatase and tensin (PTEN), resulting in remodeling of the airways (Figure 1a) [34]. The above evidence suggests that miR-21-5p is involved in the inflammatory state and the remodeling of airway epithelial cells through the induction of the epithelial–mesenchymal transition, which is an important factor for the development and exacerbation of asthma. In contrast, an OVA-induced asthma model in miR-21 knockout (KO) mice showed that the lack of miR-21 led to a decrease in airway hyperresponsiveness and eosinophilic inflammation and a reduction in Th2 cell cytokines (IL-4, IL-5, and IL-13) in the bronchoalveolar lavage fluid (BALF), along with an increase in the levels of IL-12 and IFN-γ due to the suppression of the polarization of M2 macrophages (Figure 1c) [35]. A recent study conducted in an asthmatic mouse model induced by OVA found the up-regulation of miR-21-5p in airway epithelial cells compared to control mice [36]. Likewise, the same study reported that the transfer of miR-21-5p-enriched exosomes of mast cells to asthmatic mice induced a state of oxidative stress and inflammation in epithelial cells of the airway as well as exacerbation of asthmatic symptoms by the negative regulation of dimethylarginine dimethylaminohydrolase 1 (DDAH1), resulting in the inhibition of Wnt/β-catenin signaling activity (Figure 1a) [36]. Therefore, evidence indicates that miR-21-5p is involved in remodeling and inflammation in asthma pathology.

On the other hand, few clinical studies reported a relationship between miR-21-5p and vitamin D status. In aorta tissue from atherosclerotic patients with a vitamin D deficiency (<25 nm/L), the specimens presented a greater expression of miR-21-5p with a mean of 6.7-fold, while in patients with vitamin D sufficiency (>25 nm/L), the tissue presented a lower expression of miR-21-5p with a mean of 1.7-fold [37]. Another study in diabetic type 1 patients found that diabetic patients with vitamin D deficiency (>25 nm/L) presented an increase in circulating levels of miR-21-5p, while non-diabetic patients with both sufficiency and insufficiency of vitamin D presented a decrease in circulating levels of miR-21-5p [38]. The above evidence suggests that miR-21-5p could be regulated by vitamin D. A recent study found that in TGFβ1-stimulated human bronchial fibroblast cells (asthma cell model) treated with calcitriol, the downregulation of miR-21-5p expression was induced (Figure 1a) [39]. This miRNA is related to TGFβ/Smad signaling pathway activation and fibrosis involved in the pathogenesis of pulmonary fibrosis [40]. Hence, the treatment with vitamin D_3_ has beneficial effects on asthma by inhibiting the remodeling of epithelial and fibroblasts in the lung through the regulation of miR-21 expression. Thus, the beneficial effects of vitamin D_3_ on lung tissue remodeling may be through the regulation of the expression of miR-21-5p. Thus, the beneficial effects of vitamin D_3_ on lung tissue remodeling may be through the regulation of the expression of miR-21-5p. Nevertheless, to date, few studies exist that explore the role of the treatment of vitamin D_3_ on the regulation of miR-21-5p in asthma patients as a plausible therapeutic strategy.

## 4. The Role of MicroRNA-27b in Asthma and Its Modulation by Vitamin D_3_

miR-27b-3p is a miRNA that plays an important role in regulating various physical and pathological processes, including asthma, by regulating the inflammatory response and apoptosis [41,42]. In clinical studies, for example, a study in children with mild to moderate asthma identified a negative association between levels of miR-27b-3p in serum and lung function FEV1% [24]. Moreover, a study conducted on pediatric patients with dust mite-induced asthma and dust mite-induced asthma with food allergy presented a downregulation of miR-27b-3p expression compared with children with only food allergy and healthy children [43]. It also was found that Spleen tyrosine kinase (SYK) and epidermal growth factor receptor (EGFR) regulate the development of asthma by the induction of the PI3K-AKT pathway and influence cell differentiation of T or B cells [44]. These genes are targets of miR-27b-3p, which suggests that miR-27b-3p may regulate the inflammation present in asthma by regulating the PI3K-AKT pathway [43]. However, more studies on this interaction need to be validated. In contrast, a study with patients with severe asthma showed that the treatment with anti-IL-5 promoted the downregulation of miR-27b-3p, improved lung function, and decreased peripheral eosinophil counts [17]. Furthermore, in asthmatic patients an up-regulation of miR-27b-3p in blood samples compared to healthy controls was found and it was showed, through bioinformatic analysis, that serpin family E member 1 (SERPINE1) and RAR-related orphan receptor A (RORA) are genes target of miR-27b-3p, which are genes associated with allergic asthma [45]. Likewise, human BSMC from patients with asthma presented an up-regulation of miR-27b-3p expression compared with cells from healthy controls. Through bioinformatic analysis, the same study showed that miR-27b-3p participates in abnormal activation of the phosphatase and tensin homolog (PTEN) signaling, leading to the implementation of the growth program in cells from asthmatic patients [46].

On the other hand, in experimental studies, CD4+ Th2 cells from a mouse asthma model induced by OVA challenge showed an up-regulation in both miR-23b and miR-27b expression and a downregulation in both miR-106b and miR-203 in naïve CD4+ T cells [47]. The same study demonstrated that the inhibition (miR-23b and miR-27b) and overexpression (miR-106 and miR-203) simultaneously resulted in a progressive reduction in the Th2 phenotype by the reduction in IL-5, IL-9, and IL-13 production (Figure 1c) [47]. Therefore, the evidence suggests that miR-27b-3p could be important in asthma. However, to date, few studies have shown the role of miR-27b-3p in asthma. Hence, more experimental and clinical studies are needed to elucidate the function of miR-27b-3p in the asthma context.

Few studies reported that treating vitamin D_3_ reduces remodeling in asthma by regulating miR-27b. In experimental studies, for instance, in transforming growth factor (TGF) β1-stimulated human lung fibroblasts, the expression of miR-27b induces the differentiation to myofibroblast by negative regulation of VDR (Figure 1b) [48]. Likewise, treating 1,25(OH)2D3 in these cells inhibited its differentiation by the downregulation of miR-27b [48], suggesting that the effect of vitamin D_3_ on remodeling in asthma is mediated by the downregulation of miR-27b. In contrast, epithelial and splenocyte cells reported that the treatment with vitamin D_3_ increased the expression of miR-27b-3p [49,50]. The above evidence indicates that miR-27b plays an important role in the development of asthma. Likewise, the beneficial effects of the supplementation with vitamin D_3_ are mediated by the negative regulation of miR-27b. Hence, more studies need to elucidate the role of vitamin D_3_ on the regulation of miR-27b in improving asthma.

## 5. The Role of MicroRNA-145 in Asthma and Its Modulation by Vitamin D_3_

miR-145-5p is a member of the miR-143/145 cluster, which is involved in lipid metabolism and inflammatory pathways in a tissue-specific manner [51]. Moreover, miR-145-5p is a miRNA mainly considered a tumor suppressor in several cancers. However, it has also been shown to affect the pathogenesis of several diseases, including asthma [52]. Clinical studies have shown that asthmatic patients present high plasma levels of miR-145-5p compared with healthy controls [23,53] and that high levels of this miRNA correspond to a high count of eosinophils in blood [23]. A cross-sectional study conducted in children with several asthma phenotypes showed that the levels of miR-145-5p in exhaled breath condensate correlates positively with asthma [54]. Hence, these results could suggest that miR-145-5p may play a role in the development of asthma. In contrast, a study with asthmatic patients found low levels of miR-145-5p in plasma compared with healthy controls [23]. Moreover, a study conducted in children with asthma with reduced FEV1 values (both reduced growth and early decline) presented a high reduction in miR-145-5p levels in plasma compared with both children with only reduced growth and children with normal growth [55], while children with early decline showed an increase in miR-145-5p levels compared with children with reduced growth [55]. The same study showed that children who ended follow-up (when they were aged 23–30 years) developed COPD and those subjects presented a substantial reduction in levels of miR-145-5p in plasma compared with those whose non-developed COPD [55].

Experimental models of asthma have studied the role of miR-145-5p in asthma pathogenesis. Mouse models of asthma induced by challenge with house dust mite (HDM) found a strong up-regulation of miR-145-5p in the airway wall [56]. The same study also showed that the inhibition of miR-145-5p in asthmatic mice significantly reduced the development of airway hyperresponsiveness (reduced the number of both mucus-producing cells and eosinophils present in the airways) by the reduction in the production of IL-15 and IL-13 from antigen-specific Th2 cells; these effects were comparable with dexamethasone treatment [56]. Another study with a mouse model of asthma induced by HDM found an up-regulation of miR-145-5p expression and a down-regulation of kinesin Family Member 3A (KIF3A) in airway epithelial cells [57]. The same study found that the overexpression of miR-145-5p in bronchial epithelial cells promoted the secretion of chemokines and inflammatory factors and the dysfunction of the epithelial barrier by down-regulation of KIF3A [57]; its inhibition resulted in the improvement of symptoms and the inflammatory state due to an increase in the expression of KIF3A (Figure 2a) [57]. In contrast, an asthma model in mice induced by OVA showed that the expression of miR-145-5p led to less airway remodeling by regulation of EGFR compared with asthmatic mice with inhibited miR-145-5p [58]. Likewise, the negative regulation of EGFR by miR-145-5p promoted a reduction in both the proportion of Th2 and Th17 cells in blood and inflammatory factors in asthmatic mice compared with asthmatic mice with miR-145-5p inhibited [58]. Thus, in both human and murine models, evidence indicates that miR-145-5p participates in hallmark features of allergic airway disease.

At the cellular level, human ASM cells treated with TNF-α, IL-1β, and IFN-γ (imitate the inflammatory conditions in patients with asthma) induced an up-regulation of miR-145-5p and a down-regulation of Krüppel-like factor 4 (KLF4), an inhibitor of smooth muscle cell proliferation [59]. The same study showed that the negative regulation of KLF4 by overexpression of miR-145-5p enhanced the proliferation and migration of ASM cells in vitro [59], suggesting that this miRNA may participate in the smooth muscle remodeling present in asthma pathology (Figure 2a). Moreover, Qiu Yu-Ying and collaborators found that CD4+ T cells from patients with asthma presented both an up-regulation of miR-145-5p and a down-regulation of Runt-related transcription factor 3 (RUNX3) expression compared with healthy controls [60]. Given that RUNX3 plays a role in the maintenance of Th1/Th2 balance, they conducted an individual and simultaneous inhibition of miRNAs (miR-371, miR-138, miR-544, miR-145, and miR-214) and found that only simultaneous inhibition of miRNAs increased the mRNA and protein of RUNX3 [60]. Regarding the above study, Fan Linxia and collaborators showed that T CD4+ cells from patients with asthma presented an up-regulation in both miR-145-5p expression and IL-4 (Th2) expression and a down-regulation in both IFN-γ (Th1) and RUNX3 (Figure 2b) [61]. The inhibition of miR-145-5p in T CD4+ cells from patients with asthma promoted an enhanced percentage of IFN-γ+ CD4+ T cells by regulation of RUNX3 [61]; thus, miR-145-5p can modulate the Th1/th2 balance in asthma (Figure 2b). Therefore, all evidence suggests that miR-145-5p is an important miRNA in asthma development by regulating Th1/th2 balance and airway remodeling. However, to date, no clinical trials are driven to miR-145-5p regulation for asthma treatment.

Few studies reported a relationship between miR-145-5p and vitamin D status. In clinical studies, for instance, a recent study showed that obese and non-obese type 2 diabetes mellitus (T2DM) with deficiencies of vitamin D presented an up-regulation of miR-145-5p in plasma (5.91-fold) compared with those with insufficiency (4.61-fold) and sufficiency (4.29-fold) [62]. Moreover, recently, it was demonstrated that miR-145-5p is regulated by the vitamin D_3_ treatment in gastric cancer cells [63] and in aortic tissue [64]. Evidence suggests that the supplementation of vitamin D_3_ may induce the reduction in expression of miR-145-5p, improving asthma status. However, to date, there are no studies that indicate the above. Therefore, it is necessary to evaluate the effect of the supplementation of vitamin D_3_ on the expression regulation of miR-145-5p in asthma.

## 6. The Role of MicroRNA-146a in Asthma and Its Modulation by Vitamin D_3_

miR-146a-5p is a miRNA that plays an important role in immune responses in various diseases, including asthma, regulating the immune response by suppressing the toll-like receptor 4 (TLR4)-mediated nuclear factor kappa B (NF-kB) signaling pathway [65]. In clinical studies, for instance, a study with patients with asthma found high plasma levels of miR-146a-5p compared to healthy controls. Also, it was shown that miR-146a-5p levels correlated negatively with blood eosinophil counts, higher asthma control questionnaire scores, and daily inhaled corticosteroids [66]. Similar results in exhaled breath condensate from children with several asthma phenotypes were found [54] and miR-146a-5p levels were positively associated with higher small airway reversibility [54]. Asthmatic patients found high plasmatic levels of miR-146a-5p compared with non-allergic asthma patients [67]. Likewise, miR-146a-5p levels correlated negatively with lung function (FEV1/FVC) and neutrophil count [67]. A recent study with asthmatic patients (mild and moderate–severe asthma) showed that patients with mild asthma presented high miR-146a-5p levels in exosomes in plasma compared with patients with moderate–severe asthma [29]. The same report showed that those patients with high IL-6 levels showed high miR-146a-5p levels in exosomes [29] and also reported that miR-146a-5p levels in exosomes were positively associated with IgE levels, while they were negatively associated with IL-6 levels [29,57]. Additionally, a recent study conducted on children with bronchial asthma showed that mild to moderate asthma children presented high plasma levels of miR-146a-5p compared with severe asthma children. The same study showed that miR-146a-5p levels positively correlated with Treg% [68]. A previous study reported that miR-146a-5p is required for Treg cell activity to suppress Th1-associated inflammation [69]. Hence, the above study using a logistic regression analysis found that miR-146a-5p could be considered a protective factor for developing asthma [68]. This evidence suggests that miR-146a-5p may participate as a protective factor for the development of asthma. In contrast, a study with asthmatic children showed high plasma levels of miR-146a-5p, which was positively associated with both total IgE and pulmonary function (pre-FEV1 and post-FEV1) and negatively associated with age [70]. Therefore, more studies are needed to elucidate the role of miR-146a in the development of allergic asthma.

On the other hand, in both experimental studies in animals and cellular asthma models, the inflammatory role of miR-146a-5p has been demonstrated. The coadministration of the miR-146a-5p mimic in a mouse asthma model promoted the decrease in AHR (increase in both eosinophil, neutrophil, macrophage, and lymphocyte infiltration and inflammatory mediators, namely OVA-specific IgE and Th2 cytokines: IL-4, IL-5, and IL-13 in BALF) by the suppression of ILC2 responses [71]. These effects of miR-146a-5p were dependent on IL-33 stimulation (Figure 3b) [71]. The knockout MIR146a/b^−/−^ mouse asthma model induced by HDM showed an increased number of B cells and strongly reduced numbers of Th2 cells in BALF compared with wild asthmatic mice [72]. Instead, the same knockout MIR146a/b^−/−^ mouse asthma model infected with rhinovirus infection for the exacerbation of asthma identified a robust increase in neutrophils and an increase in Th1 and Th17 cells, FOXP3+ regulatory T cells (Tregs), and CD44+ memory T cells in comparison with knockout MIR146a/b^−/−^ asthmatic mice and wild asthmatic mice [72]. Moreover, a mouse asthma model with the systemic administration of small extracellular vesicles enriched with miR-146a-5p from human mesenchymal stromal cells exhibited alleviation of airway hyperresponsiveness (decreasing inflammatory cell infiltration and mucus production in the lung and a reduction in levels of Th2 cytokines) by inhibition of ILC2 levels [73].

At the cellular level, airway epithelial cells from patients with eosinophilic, neutrophilic, and paucigranulocytic asthma phenotype presented a downregulation of miR-146a-5p expression related to healthy controls [74] and a negative correlation between miR-146a-5p expression and neutrophil counts in BALF from patients with asthma was found [74]. The same study showed that the overexpression of miR-146a-5p in human bronchial epithelial cells promoted the reduction in neutrophil migration and IL-18 and C-X-C motif ligand 1 (CXCL1) secretion [74]. Human lung epithelial cells stimulated with IFN-γ, IL-1β, and TNF-α to simulate the inflammatory state present in asthma showed an up-regulation of miR-146a-5p along with an increase in 5-lipoxygenase (5-LO) activity [75]. Moreover, inhibiting miR-146a-5p in the same cells promoted an intense activity of 5-LO [75]. These results suggest that miR-146a-5p may modulate the formation of pro-inflammatory leukotrienes by regulating 5-LO activity. In human small airway epithelial cells (HSAECs) stimulated with platelet-activating factor (PAF), an inductor of the inflammatory response, downregulation of miR-146a-5p expression was promoted [76], while the overexpression of miR-146a-5p in PAF-stimulated HSAECs induced a reduction in airway inflammation (IL-1β, Il-6, and TNF-α) and cell barrier damage (apoptosis) in asthma by targeting TNF receptor-associated factor 6 (TRAF6) (Figure 3a) [76]. A recent study showed in macrophages from asthmatic mice that miR-146a-5p blocked M1 polarization by inhibiting the toll-interleukin-1 receptor domain-containing adaptor protein (TIRAP)/NF-κB pathway, resulting in M2 polarization promotion [77]. These results suggest that miR-146a-5p regulates the inflammatory response present in asthma by different mechanisms. Opposite results were found by Zhang Y and collaborators, who showed that the overexpression of miR-146a-5p in human bronchial smooth muscle cells (BSMC) inhibited the proliferation cellular and induced apoptosis by targeting EGFR (Figure 3a) [78]. Furthermore, in human bronchial epithelial cells stimulated with TNF-α, the up-regulation of miR-146a-5p was induced along with an increase in the expression of inflammatory cytokines (CCL2, CCL5, IL-6, GM-CSF, CXCL-1, and IL-8) [66]. The same study reported that the overexpression of miR-146a-5p in human bronchial epithelial cells stimulated with TNF-α attenuated the expression of inflammatory cytokines and, when it combined miR-146a-5p and dexamethasone (DEX), it provided better inhibition of cytokine production in compared with DEX alone [66]. Therefore, all evidence suggests that miR-146a has an important inflammatory role in asthma by the regulation of balance in Th1/Th2 cells.

Recent experimental studies reported that vitamin D_3_ treatment can regulate the expression of miR-146a-5p. Vitamin D_3_ treatment induced the up-regulation of miR-146a-5p in hepatic stellate cells [79] and skin wound healing in diabetic mice [80], improving biological status. The treatment with vitamin D_3_ had an anti-inflammatory effect in both TNF-α-stimulated adipocytes and the adipose tissue of an inflammatory mouse model by the reduction in miR-146a-5p expression [81]. This vitamin D_3_ effect was through the regulation of NF-κB signaling. In contrast, the stimulation with vitamin D_3_ in dengue virus type 2 (DENV-2)-infected human macrophages induced the downregulation of miR-146a-5p expression [82], as well as in an infarct mouse model with aerobic-resistance training where the supplementation with vitamin D_3_ promoted a downregulation of circulating miR-146a-5p [83]. The evidence suggests that the vitamin D_3_ supplementation may regulate the inflammation present in asthma by the regulation of miR-146a-5p expression. However, no studies evaluate the above in the asthma context. Because of the crucial role of miR-146a-5p as a protective factor in the development of asthma, it is important to assess whether the supplementation of vitamin D_3_ may up-regulate miR-146a-5p expression.

## 7. The Role of MicroRNA-155 in Asthma and Its Modulation by Vitamin D_3_

MiR-155 is encoded by the miR host gene 155 (MIRHG155), identified initially as the B-cell integration cluster gene, which is a miRNA important for the immune response and is involved in several diseases, including asthma. A clinical study conducted on asthmatic children reported that those children presented high plasmatic levels of miR-155-5p compared to healthy children, which was associated with an increased risk of childhood asthma [84]. It is well known that indoor air pollution aggravates allergic asthma. The same study reported that the plasmatic levels of miR-155-5p positively correlated with particulate matter 2.5 and formaldehyde [84]. A recent study found that asthmatic patients presented high serum levels of miR-155-5p compared to healthy subjects [30]. Also, only asthma presence in these patients was a significant predictor of miR-155-5p expression [30]. A study conducted in asthmatic children reported that those children presented an up-regulation of miR-155-5p in plasma comparisons with healthy children, which increased with disease severity, along with increased IL-13 levels [85]. Furthermore, the same study reported that plasma levels of miR-155-5p positively correlated with plasma levels of IL-13 and negatively with lung functions (FEV1 and FVC) [85]. Therefore, evidence suggests that altered expression of miR-155-5p plays an important role in the development and exacerbation of asthma.

In experimental animals, miR-155-p has been found to participate in the inflammatory response of airway hyperresponsiveness and allergic airway inflammation. For instance, the lung tissue from a mouse asthma model found the up-regulation of miR-155-5p compared to control mice [86]. Likewise, the same study demonstrated that silencing of miR-155-5p reduced the levels of cytokines IL-4, IL-5, and IL-13 and the total eosinophil, macrophage, and lymphocyte count in BALF from asthmatic mice [86]. According to the above study, in a miR-155 KO mouse model of asthma, it was shown that a lack of miR-155 promotes a reduction in Th2 cells number and cytokines IL-4, IL-5, and IL-13, resulting in the decrease in eosinophil-mediated inflammation and mucus hypersecretion in the lung compared to wild-type asthmatic mice (Figure 2b) [87]. In another miR-155 KO mouse asthma model, it was observed that the lack of miR-155 promotes the reduction in reactive species oxygen (ROS) and cyclooxygenase 2 (COX-2) levels, Th2 and Th17-associated cytokines (IL-4, IL-13, and IL-17) levels in BALF, and in eosinophil infiltration [88]; the expression of miR-155 presented opposite results in the miR-155 KO mouse asthma model (Figure 2b) [88]. Moreover, a study with IL-4- and IL-13-treated macrophages stimulated with IL-33 (an asthma cellular model) induced an increase in the expression of miR-155-5p, along with an increase in inflammatory mediators (Ccl3, Ccl5, Ccl17, Ccl24, and Il1b) compared to macrophages without pretreatment [89]. Therefore, evidence suggests that the allergic inflammatory response is potentiated through the increase in the expression of miR-155-5p.

Experimental studies have reported that treating vitamin D_3_ improves the inflammatory response by regulating miR-155 expression. A previous study reported that LPS-pretreated macrophages stimulated with 1,25(OH)2D3 induced the downregulation of the expression of miR-155-5p via blocking NF-κB activation [90]. Moreover, it was also shown that 1,25(OH)2D3 treatment up-regulates SOCS1 by decreasing miR-155-5p expression [90]. Given that SOCS1 has an important role in the negative feedback in the inflammatory response, vitamin D_3_ limits the inflammatory response by promoting negative feedback action by downregulating miR-155-5p. DENV-infected monocyte-derived macrophages (MDM) treated with 1,25(OH)2D3 promoted the downregulation of TLR4 and reduced NF-kB activity, resulting in the decrease in miR-155-5p expression and IL-1β and TNFα production, along with the up-regulation of SOCS1 expression compared with DENV-infected cells not treated with MDM [91]. Another study in 1,25(OH)2D3-pretreated adipocytes stimulated with TNFα showed a decreased expression of miR-155-5p compared with non-pretreated adipocytes [81]. This downregulation of miR-155-5p was due to the pretreatment with 1,25(OH)2D3 reduced NF-κB activation by reducing phosphorylation levels of p65 and IκB [81]. Hence, the above evidence suggests that vitamin D_3_ supplementation could regulate the inflammatory response through the regulation of miR-155-5p. However, little is known about the beneficial effects of vitamin D_3_ on miR-155-5p expression in asthmatic patients. Therefore, studies to elucidate the anti-inflammatory effect of vitamin D_3_ through the regulation of miR-155 in asthma are needed.

## 8. The Role of Miscellaneous MicroRNAs and Vitamin D_3_ in Asthma

A recent clinical study reported that vitamin D_3_ improves asthma by regulating miRNAs. For instance, in a vitamin D antenatal asthma reduction trial (VDAART) conducted by Li Jiang and collaborators, children were divided into two groups: (1) children born to mothers who had received 4400 IU of vitamin D_3_ per day during pregnancy and (2) children born to mothers who had received regular multivitamins containing 400 IU of vitamin D_3_ per day (as placebo group). In the group with high vitamin D_3_ treatment, it was found that 32 miRNAs were associated with asthma, of which 22 were associated with a high risk of incident asthma and 10 miRNAs were protective [20], while in the group with low vitamin D_3_ treatment, 6 miRNAs associated with asthma were found, of which miR-505-3p and miR-340-3p had a high risk of incident asthma [20]. The same study performed a meta-analysis of the results of the VDAART cohort with the results of children from a prenatal cohort (VIVA PROJECT, same study conditions). This study found that in the high vitamin D_3_ treatment group, the miR-574-5p has a great risk for incident asthma and miR-151a-5p has the highest protective effect [20]. Regarding miR-151a, a recent study showed that in T cells, the expression of miR-151a can negatively regulate Th1 cytokine expression (IL-2, IL-12, and IFN-γ) through the regulation of interleukin-12 receptor β2 (IL12RB2) [92]. Moreover, patients with severe asthma presented dominant Th1-associated cytokine expression profiles and neutrophilic inflammation [93,94,95] and a negative correlation between miR-151a levels and neutrophilic inflammation was identified in asthmatic patients [96]. Therefore, the above evidence suggests that the supplementation with vitamin D_3_ may improve asthma status, in part, by the regulation of miR-151a expression.

On the other hand, in allergic reactions, the supplementation with vitamin D_3_ in allergic rhinitis patients induced the downregulation of miR-19a in B cells [97]. Also, it was found that B cells can hydroxylate to vitamin D_3_, forming complex vitamin D_3_/VDR/RXR (CVR), which binds the miR-19a promoter to interfere with the gene transcription of miR-19a [97]. Moreover, LPS/IL-4-stimulated miR-19a-deficient B cells promoted an increase in IL-10 production, improving the allergic status [97]. A previous study reported that bronchial epithelial cells from severe asthmatic patients presented an up-regulation of miR-19a compared to healthy subjects. The same study also found that miR-19a enhances cell proliferation of bronchial epithelial cells by targeting the TGF-β receptor 2 [98]. A recent study showed that the inhibition of miR-19a in a mouse asthma model attenuated inflammation and mucus production induced by Th1 cells, suppressed the Th2 inflammatory response, and repressed dendritic cell maturation by increasing RUNX3 [99]. The same study showed that miR-19a induces Th1 polarization and inhibits Th2 polarization by directly acting on naïve CD4+ T cells by targeting RUNX3. The above evidence suggests that miR-19a is involved in remodeling epithelial cells and the inflammatory response of Th2 cells [99]. Therefore, the vitamin D_3_ treatment may reduce the lung tissue remodeling and inflammation present in asthma by regulating the expression of miR-19a. Nevertheless, there are no studies about the effect of vitamin D_3_ treatment on miR-19a expression in asthma.

In dengue lung infection, a recent study demonstrated that the vitamin D_3_ treatment induces an increase in the miR-34a expression in monocyte-derived macrophages [100]. Hence, miR-34a could be a miRNA regulated by vitamin D_3_ activity. Emerging studies suggest a role of the miR-34a in asthma, for instance, patients with moderate asthma showed a decrease in serum levels of miR-34a in comparison with both obese and healthy subjects [101]. The same study demonstrated, through an in-silico analysis, that miR-34a can regulate several genes related to asthma and lung injury [101]. Likewise, in children with mild-to-moderate and severe asthma, it was observed that miR-34a correlated negatively with function lung parameters (FEV1FVC%pred and FEF2575%pred) [102], and with proinflammatory cytokines (TNF-α, IL-1β, IL-6, and IL-17) in children with remissive asthma and exacerbated asthma [103]. Thus, miR-34a may participate in the development of asthma.

Moreover, in asthma mice models induced by OVA challenge, an increase in the expression of miR-34a in lung tissue was presented [104,105]. Moreover, in a lung injury mice model induced by bleomycin, the miR-34 expression was up-regulated in both fibrotic lung and myofibroblasts [106]. Likewise, in mice knockout for miR-34, an inhibition in both senesces and apoptosis in lung fibroblast was shown [106]. At the cellular level, in alveolar epithelial cells from lung injury, the mice model showed an increase in miR-34a expression [107]. The same study found that the expression of miR-34a induced apoptosis by increasing the p53 acetylation through regulation of sirtuin 1 (SIRT1) [107]. Furthermore, in airway smooth muscle cells, the expression of miR-34a reduces proliferation and migration by regulation of vesicle-associated membrane protein 2 (VAMP2) [108]. Thus, evidence suggest that miR-34 has a protector role in asthma and lung injury and that vitamin D_3_ supplementation may reduce the lung injury present in asthma by the regulation of miR-34a.

Recent research has highlighted the modulation of miR-181a through Vitamin D_3_ treatment. Specifically, it has been observed that the treatment of peripheral blood mononuclear cells with vitamin D_3_ leads to an upregulation in the expression of miR-181a [109], suggesting that vitamin D_3_ could regulate this miRNA. A recent study showed that patients with moderate asthma showed a decrease in serum levels of miR-181a [101]. Through in silico analysis, it was found that miR-181a could regulate genes related to the exacerbation of asthma and respiratory failure [101]. Likewise, in children with exacerbated and controlled asthma, miR-181a expression negatively correlated with serum levels of TNF-α, IL-1β, and IL-6 [110]. This evidence suggests the important role of miR-181a in the development of asthma.

On the other hand, in allergic rhinitis mice models, a recent study showed decreases in the expression of miR-181a in both nasal mucosal and lung tissue [111]. Also, it was found that in serum, the expression of miR-181a decreases the levels of OVA-specific IgE, along with diminished IL-6 and IFN-γ (Th17) and increased IL-10 (Treg) [111]. Interestingly, the expression of miR-181a in the lung of allergic rhinitis mice reduced the count of eosinophils in BALF and decreased IL-4, IL-5, and IL-13 by targeting high mobility group box 1 (HMGB1), resulting in retarded development of asthma [111]. Moreover, an allergic rhinitis model induced by OVA in human nasal epithelial cells showed a decrease in the miR-181a expression and its overexpression reduces the production of TNF-α, IL-1β, and IL-6 through the regulation of IL-33 signaling [110]. Therefore, the treatment with vitamin D_3_ may regulate the inflammatory response present in asthma through the up-regulation of miR-181a expression.

## 9. Future Directions

More studies that assess the vitamin D_3_ effects on the regulation of miRNAs related to inflammation in asthmatic patients are needed;The impact of genetic variants on vitamin D receptors on the regulation of microRNAs associated with asthma should be assessed;The effect of the intake of vitamin D_3_-enriched foods vs. the supplementation of vitamin D_3_ on the regulation of miRNA expression in asthmatic patients should be assessed.

## 10. Conclusions

In conclusion, miRNAs play an important role in the development and exacerbation of asthma. From emerging studies using the supplementation of vitamin D_3_, it has been shown that vitamin D_3_ can improve asthma symptoms. Evidence suggests that supplementation with vitamin D_3_ improves asthma by regulating miRNAs. Since miR-21, miR-27b, miR-145, miR-146a, and miR-155 are related to the disbalance of Th1/Th2 cells, inflammation, and airway remodeling, which worsens asthma symptoms, the effect of vitamin D_3_ on the regulation of these miRNAs in asthma must therefore be elucidated as a therapeutic strategy.

## Figures and Tables

**Figure 1 nutrients-16-00341-f001:**
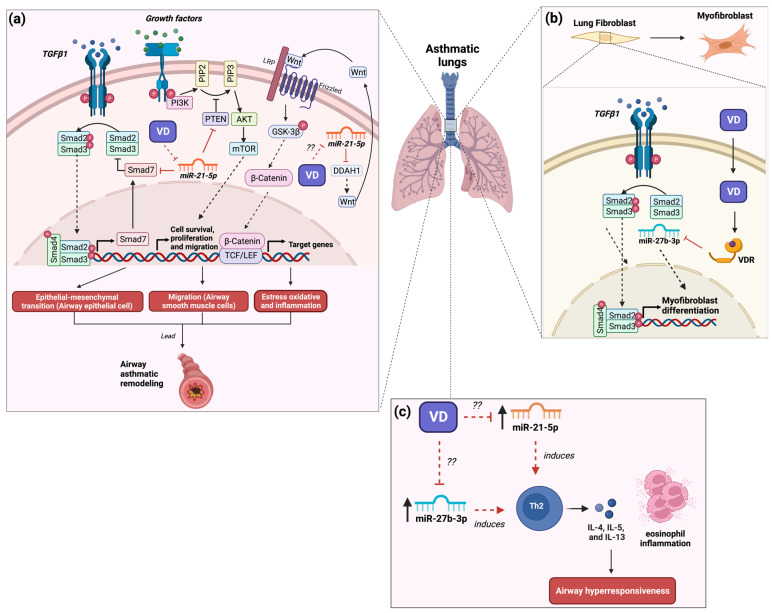
Role of miR-21 and miR-27b in the development of asthma and the possible implication of vitamin D_3_ supplementation. (**a**) The expression of miR-21-5p regulates signaling pathways such as TGFβ, mTOR, and Wnt/β-catenin involved in airway remodeling and inflammation. Vitamin D_3_ supplementation may decrease the expression of miR-21-5p, improving asthma symptoms by reducing TGFβ and mTOR activity and increasing β-catenin activity. The question marks indicate a possible effect of vitamin D3 on oxidative stress and inflammation through the regulation of miR-21. (**b**) The expression of miR-27b-3p in lung tissue promotes fibroblast differentiation to myofibroblast. The interaction between vitamin D_3_ and VDR reduces the expression of miR-27b-3p, inhibiting the differentiation of lung fibroblasts. (**c**) The expression of miR-21-5p and miR-27b-3p induce Th2 differentiation, IL-4, IL-5, and IL-13 release, and eosinophil inflammation, resulting in airway hyperresponsiveness. The question marks suggest a possible effect of vitamin D3 on airway hyperresponsiveness via the regulation of miR-21 and miR-27b, which could result in the reduction of Th2 cell levels. Studies showed that vitamin D_3_ supplementation improves airway hyperresponsiveness, which may be by regulating these miRNAs [37]. The red lines illustrate the regulatory effects of both vitamin D3 and miRNAs. Created with BioRender.com.

**Figure 2 nutrients-16-00341-f002:**
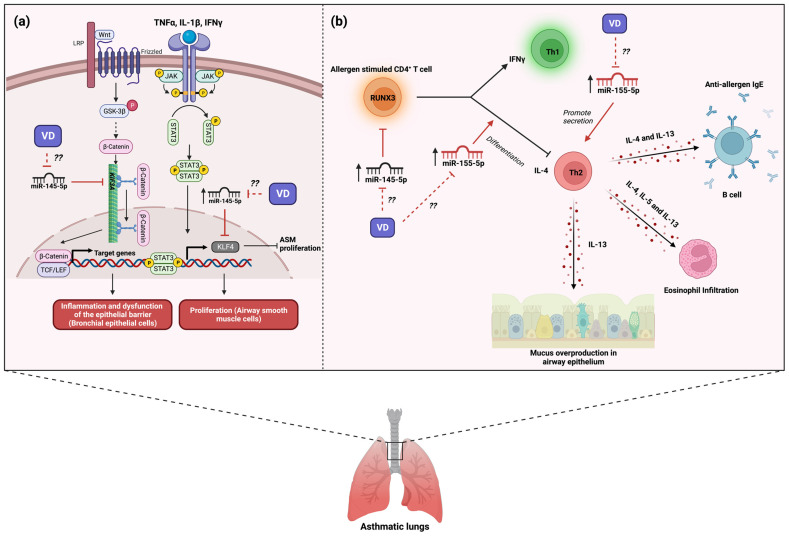
Role of miR-145-5p and miR-155-5p in the development of asthma and possible implication of vitamin D_3_ supplementation. (**a**) The expression of miR-145-5p in asthma is involved in the interference of the Wnt/β-catenin and cytokine signaling, leading to inflammation, dysfunction of bronchial epithelium, and the proliferation of airway smooth muscle. The question marks indicate that Vitamin D_3_ supplementation may alleviate the inflammation, avoiding airway epithelium dysfunction and smooth muscle proliferation by inhibiting miR-145-5p expression. (**b**) Likewise, both miR-145-5p and miR-155-5p induce Th2 differentiation, promoting the Th2 phenotype. The expression of miR-155-5p promotes the secretion of Th2-associated cytokines, which induce mucus overproduction, eosinophil infiltration, and anti-allergen IgE production. The question marks indicate that vitamin D_3_ supplementation may modulate the balance of Th1/Th2 cells through the regulation of miR-145-5p and miR-155-5p expression, potentially leading to the alleviation of asthma symptoms. The red lines illustrate the regulatory effects of both vitamin D3 and miRNAs. Created with BioRender.com.

**Figure 3 nutrients-16-00341-f003:**
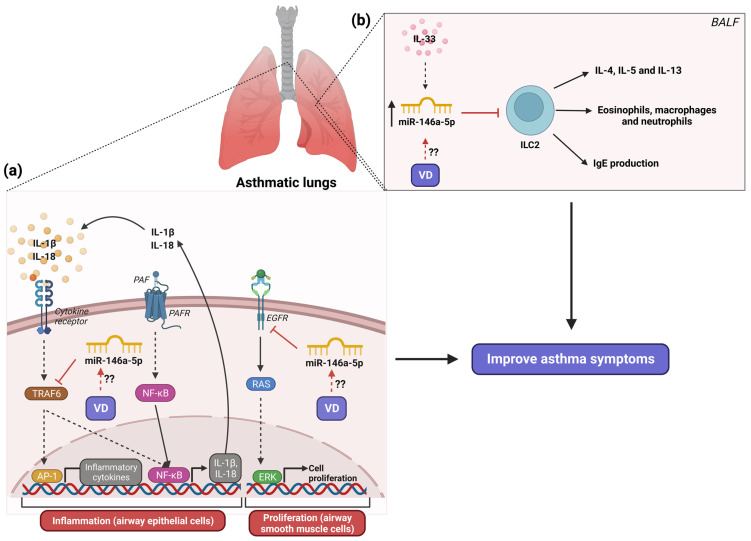
Role of miR-146a-5p in the development of asthma and the possible implication of vitamin D_3_ supplementation. (**a**) In asthma, miR-146a-5p acts as a protector factor that improves the inflammatory response and airway smooth muscle proliferation by regulating signaling pathways of cytokines, PAF, and EGFR. (**b**) Likewise, miR-146a-5p inhibits ILC2 activation, thus decreasing IL-4, IL-5, and IL-13 release, eosinophils, macrophages, and neutrophils infiltration and IgE production. The question marks indicate that Vitamin D_3_ supplementation may increase the expression of miR-146a-5p, causing the amplified function of this miRNA, which results in improved asthma symptoms. The red lines illustrate the regulatory effects of both vitamin D_3_ and miRNAs. Created with BioRender.com.

## Data Availability

Not applicable.

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
