# Peer review of "The Roles of MicroRNAs in Asthma and Emerging Insights into the Effects of Vitamin D3 Supplementation"

_nutrients, 2024, doi:10.3390/nu16030341_

Round 1
Reviewer 1 Report
Comments and Suggestions for Authors
This review addresses the molecular mechanisms of asthma and how microRNAs affect the development and its exacerbation, and how vitamin D3 may modulate microRNAs to improve asthma symptoms. This review is comprehensive, but there are some deficiencies:
1. The title of this review is “Possible Effects of the Supplementation with Vitamin D3 on microRNAs Expression in Patients with Asthma”, the authors described the mechanism of microRNAs and asthma, but when it comes to the effect of vitamin D3 supplementation, the description is always too little due to the lack of studies, so I think this is inconsistent with the topic of this paper.
2. Introduction: this chapter only mentioned vitamin D at the end, and then decided to address epidemiological studies of the relationship between asthma and vitamin D deficiency/insufficiency and the molecular mechanisms mediated by microRNAs. I think it's a little abrupt unless you change the chapter name and reorganize the section.
3. Line 97, the first appearance of FEV and FVC in this review does not have the full name and the meaning of the indicator, and non-professionals cannot accurately understand.
4. Line 145-148, you mentioned that moderate-severe asthma patients presented elevated levels of miR-21-5p compared with both mild asthma and healthy subjects, which means higher miR-21-5p is harmful. But at line 148-149, the same study reported the levels of miR-21-5p correlated positively with FEV1/FVC, which showed lowed levels in asthma patients. This is confusing to me, so please explain.
5. Line 158-162, references are lacked.
6. Part 4: Few studies reported the relationship between miR-21-5p and vitamin D status, which determines this part of the content is also weak and can be used as an innovative experiment, but is it appropriate to appear in a review article?
7. Part 5: same with Part 4.
8. Part 6: same with Part 4.
9. Part 7: same with Part 4.
10. Part8: same with Part 4.
Comments on the Quality of English LanguageThe whole quality of English language is commendable, however, attention should be paid to certain details such as presenting the full name upon first appearance and maintaining consistentency in formatting throughout the entire text, for instance, 1-18% and 25%-75%.
Author Response
- The title of this review is “Possible Effects of the Supplementation with Vitamin D3 on microRNAs Expression in Patients with Asthma”, the authors described the mechanism of microRNAs and asthma, but when it comes to the effect of vitamin D3 supplementation, the description is always too little due to the lack of studies, so I think this is inconsistent with the topic of this paper.
R= Thank you for your observations. In line with your commentary, the manuscript title has been revised. It has been modified from “Possible Effects of the Supplementation with Vitamin D3 on microRNAs Expression in Patients with Asthma” to “The Roles of MicroRNAs in Asthma and Emerging Insights into the Effects of Vitamin D3 Supplementation.”
- Introduction: this chapter only mentioned vitamin D at the end, and then decided to address epidemiological studies of the relationship between asthma and vitamin D deficiency/insufficiency and the molecular mechanisms mediated by microRNAs. I think it's a little abrupt unless you change the chapter name and reorganize the section.
R= Thank you for your observations. In line with your suggestion, the introduction section has been thoroughly revised and modified to enhance the flow and clarity of the information presented. This revision was facilitated by a change in the title of the manuscript, ensuring better alignment with the content.
“Asthma is one of the most common chronic non-communicable diseases worldwide, affecting 1–18% of the populations of different countries [1]. In 2017, asthma incidence worldwide was 43.13 million new cases/year, while the prevalence was 272.68 million cases and mortality of 0.49 million deaths [2]. Moreover, asthma prevalence is higher in industrialized countries than in low-income and middle-income countries [1] For instance, in the United States, asthma prevalence increased from 20 million (7.3%) to 25 million (8.0%) from 2001 to 2017 [3]. Likewise, asthma prevalence was slightly higher in US children (8.4%) than in US adults (7.7%) [3]. In LATAM, findings the International Study of Asthma and Allergies in Childhood reported an asthma prevalence of 18% among children aged 13 to 14 years [4]. While higher-income nations tend to have greater asthma prevalence rates, it's noteworthy that the majority of asthma-related deaths occur in lower to middle-income regions like LATAM, where the asthma mortality rate stands at 26.3 per 100,000 individuals [4].
Asthma is a common chronic airway disease characterized by variable airflow limitation secondary to airway narrowing, airway wall thickening, and increased mucus [5], resulting from chronic inflammation and airway remodeling [5,6]. These alterations lead to airway hyperresponsiveness (AHR), airway obstruction, airflow limitation, and progressive decline of patients’ lung function [7]. Asthma is a heterogeneous disease with several distinct clinical presentations (phenotypes) and complex pathophysiological mechanisms (endotypes) [6]. Based on endotypes, asthma can be categorized into diverse classifications, including non-allergic and allergic, non-eosinophilic and eosinophilic, as well as type 2 (T2) high and T2 low, or its equivalent non-T2, with respect to the inflammatory profile [6]. In the T2 inflammation is involving of the innate (type 2 innate lymphoid cell (ILC2)) and adaptative (T-helper type 2 cells (Th2)) immune system [8]. When ILC2 and Th2 cells are triggered by contact with an allergen, they produce the type-2 cytokines (interleukin (IL)-4, IL-5, and IL-13). IL-4 and IL-13 induce B cell class switching and IgE production, the release of pro-inflammatory mediators, barrier disruption, and tissue remodeling, and IL-13 induces goblet-cell hyperplasia and mucus production [9]. Altogether, these cytokines recruit eosinophils to tissues, generating clinical symptoms of chronic inflammatory airway diseases [9]. This complex pathophysiology coupled with the global burden of asthma, underscores the urgent need for the exploration of novel therapeutic strategies. Among these, the potential role of Vitamin D3 in the management of asthma presents a promising avenue for exploration.
Vitamin D3 [1,25-dihydroxy vitamin D (1,25(OH)2D); also called calcitriol] may offer therapeutic benefits in asthma through multifaceted mechanisms. Its anti-inflammatory properties encompass the modulation of immune responses, particularly the regulation of pro-inflammatory cytokines linked to asthma through Vitamin D receptor (VDR) signaling [10,11]. Recent epidemiological studies have reported a relationship between Vitamin D deficiency/insufficiency and asthma [12–15] and supplementing of Vitamin D3 have beneficial effects on the development and exacerbation of asthma [16,17]. Despite several advances in the field, the molecular mechanisms related to the beneficial effects of vitamin D in asthma are little Notably, Vitamin D3's impact on microRNA (miRNA) expression, highlights its role in regulating molecular key pathways associated with asthma. known.
In this sense, miRNAs represent a plausible molecular mechanism through which Vitamin D3 exerts its beneficial effects on the inflammatory response. These small, non-coding RNAs are pivotal in post-transcriptional gene regulation [18] and have been increasingly recognized for their role in modulating immune responses [19]. Notably, Vitamin D3 supplementation has been observed to alter the miRNA expression profile in both plasma and cells [20,21]. This modulation of miRNA expression is particularly relevant in the context of diseases like asthma, where miRNA deregulation is a common feature.
Emerging evidence underscores that miRNAs are differentially expressed in individuals with asthma compared to those without, highlighting their significant immunoregulatory roles [22]. These miRNAs are not only cell/tissue-specific but have also been directly associated with asthmatic pathology [23–25]. Within T helper (Th) cell subpopulations, including Th1, Th2, Th17, and Treg cells, distinct miRNA signatures have been identified [26]. These signatures not only reflect the activation status of these cells but also the nature of inflammation characterizing various asthma phenotypes/endotypes and their severity levels. Targeting these miRNAs, especially through the strategic use of Vitamin D3 supplementation, could offer a novel approach to modulate the inflammatory state in asthma.” P. 1-2, L. 42-105.
- Line 97, the first appearance of FEV and FVC in this review does not have the full name and the meaning of the indicator, and non-professionals cannot accurately understand.
R= Thank you for your observations. In accordance with your suggestions, the full names for FEV (P. 3, L. 115), FVC (P. 3, L. 116), and FEF (P. 3, L. 133) have been added to enhance clarity in the manuscript.
- Line 145-148, you mentioned that moderate-severe asthma patients presented elevated levels of miR-21-5p compared with both mild asthma and healthy subjects, which means higher miR-21-5p is harmful. But at line 148-149, the same study reported the levels of miR-21-5p correlated positively with FEV1/FVC, which showed lowed levels in asthma patients. This is confusing to me, so please explain.
R= Thank you for your comments. In accordance with your suggestion, we identified an inaccuracy in the manuscript. Specifically, in the sentence "the same study reported that the levels of miR-21-5p in exosomes correlated positively with IgE levels and lung function (FEV1/FVC) and correlated negatively with plasma levels of TNF-α and IL-6 in patients with moderate-severe asthma ". We have modified the sentence to reflect this finding more accurately, based on your feedback. The revised sentence now reads: "the same study reported that the levels of miR-21-5p in exosomes correlated positively with IgE levels and correlated negatively with plasma levels of TNF-α and IL-6 in patients with moderate-severe asthma [27]". P. 4, L. 250-252.
- Line 158-162, references are lacked.
R= Thank you for your observation. In line with your commentary, the Reference 32 has been included at the specified location (P. 4, L. 167 and 169) in the manuscript.
Reference: “Joo, H., Park, S. Y., Park, S. Y., Park, S. Y., Kim, S. H., Cho, Y. S., Yoo, K. H., Jung, K. S., & Rhee, C. K. (2022). Phenotype of Asthma-COPD Overlap in COPD and Severe Asthma Cohorts. Journal of Korean medical science, 37(30), e236. https://doi.org/10.3346/jkms.2022.37.e236”
- Part 4: Few studies reported the relationship between miR-21-5p and vitamin D status, which determines this part of the content is also weak and can be used as an innovative experiment, but is it appropriate to appear in a review article?
- Part 5: same with Part 4. Part 6: same with Part 4. Part 7: same with Part 4. Part8: same with Part 4.
R= Thank you for your observation. In line with your commentary regarding the inclusion of the relationship between miR-21, miR-27b, miR-145, miR-146a and miR-155 and effect of vitamin D3 in our review. We understand your concern about the limited research in this area. Our decision to include this topic was driven by its emerging significance in the effect of Vitamin D3 in asthma through the regulation of microRNAs. While studies are few, we believe that even preliminary findings, are valuable for sparking further research and discussion in the scientific community. We have presented this information carefully, ensuring readers are aware of its preliminary nature.
Reviewer 2 Report
Comments and Suggestions for Authors
Dear Authors,
I send you my comments:
1) Please clarify the type of review
2) please add the method used to check the manuscript
3) please separate experimental studies from clinical studies
4) please why you indicated only these mir? please check these manuscripts add the data and cite in references: doi 10.3390/jcm11185446
4) why you indicate the role of mir in pain? please report only data in asthma.
Comments on the Quality of English Languagenone
Author Response
1. Please clarify the type of review
R= Thank you for your comment. To clarify, the manuscript in question is a narrative review.
2. Please add the method used to check the manuscript
R= Thank you for your observation. In response to your suggestions, it's important to note that the narrative review format prescribed by the Nutrients journal does not include a methods section.
As per the Nutrients journal's instructions for authors: Review manuscripts should comprise the front matter, literature review sections and the back matter. The template file can also be used to prepare the front and back matter of your review manuscript. It is not necessary to follow the remaining structure. Structured reviews and meta-analyses should use the same structure as research articles and ensure they conform to the PRISMA guidelines.
Based on these guidelines, the methods section was not included in our review.
3. Please separate experimental studies from clinical studies
R= Thank you for your observation. In response to your suggestion, we have now separated the clinical studies from the experimental studies for clearer distinction and better organization in the manuscript.
4. Please why you indicated only these mir? please check these manuscripts add the data and cite in references: doi 10.3390/jcm11185446
R= Thank you for your observation. In this review, the selected microRNAs were chosen specifically due to their significant roles in the development and exacerbation of asthma. They are involved in both tissue remodeling and the inflammatory response within the airways, which are crucial aspects of asthma pathology. Additionally, these microRNAs were chosen based on emerging evidence indicating that their regulation can be influenced by Vitamin D3 supplementation. Additionally, we are currently conducting a clinical study to investigate the impact of Vitamin D3 supplementation on the modulation of these specific miRNAs in the context of asthma.
Based on your suggestions, additional information has been incorporated into the “The Role of Miscellaneous MicroRNAs and Vitamin D3 in Asthma' section, detailing the other microRNAs mentioned in the article provided.
“In dengue lung infection, recent study demonstrated that the vitamin D3 treatment in-duces an increase of the miR-34a expression in monocyte-derived macrophages [101]. Hence, miR-34a could be a miRNA regulated by vitamin D3 activity. Emerging studies suggest a role of the miR-34a in asthma, for instance, patients with moderate asthma showed a decrease in serum levels of miR-34a in comparison with both obese and healthy subjects [102]. The same study demonstrated through an in-silico analysis found that miR-34a could regulate several genes related to asthma and lung injury [102]. Likewise, in children with mild-to-moderate and severe asthma was observed that miR-34a correlated negatively with function lung parameters (FEV1FVC%pred and FEF2575%pred) [103], and with proinflammatory cytokines (TNF-a, IL-1b, IL-6 and IL-17) in children with remissive asthma and exacerbated asthma [104]. Thus, miR-34a may participate in the development of asthma.
Moreover, in asthma mice models induced by OVA challenge presented an increase in the expression of miR-34a in lung tissue [105,106]. Moreover, in a lung injury mice model induced by bleomycin, the miR-34 expression was up-regulated in both fibrotic lung and myofibroblasts [107]. Likewise, in mice knockout for miR-34 showed an inhibition in both senesces and apoptosis in lung fibroblast [107]. At cellular level, in alveolar epithelial cells from lung injury mice model showed an increase of miR-34a expression [108]. The same study found that the expression of miR-34a induced apoptosis by in-creasing of the p53 acetylation through regulation of sirtuin 1 (SIRT1) [108]. Furthermore, in airway smooth muscle cells, the expression of miR-34a reduces the proliferation and migration by regulating of vesicle-associated membrane protein 2 (VAMP2) [109]. Thus, evidence suggest that miR-34 has a protector role in asthma and lung injury, and vitamin D3 supplementation may reduce the lung injury present in asthma by the regulation of miR-34a.
Recent research has highlighted on the modulation of miR-181a through Vitamin D3 treatment. Specifically, it has been observed that the treatment of peripheral blood mononuclear cells with Vitamin D3 leads to an upregulation in the expression of miR-181a [110], suggesting that vitamin D3 could regulate this miRNA. Recent study showed that in patients with moderate asthma showed a decrease in serum levels of miR-181a [102]. Through in silico analysis was found that miR-181a could regulate genes related to ex-acerbation asthma and respiratory failure [102]. Likewise, in children with exacerbated and controlled asthma, miR-181a expression negatively correlated with serum levels of TNF-a, IL-1b, IL-6 [111]. This evidence suggests the important role of miR-181a in the development of asthma.
On the other hand, in allergic rhinitis mice models, recent study showed that de-creases the expression of miR-181a in both nasal mucosal and lung tissue [112]. Also found that in serum, the expression of miR-181a decrease the levels of OVA-specific IgE, along with diminished of IL-6 and IFN-g (Th17) and increase of IL-10 (Treg) [112]. Interestingly, the expression of miR-181a in lung of allergic rhinitis mice reduced the count of eosinophils in BALF and decreased IL-4, IL-5, and IL-13 by targeting high mobility group box 1 (HMGB1), resulting in retarded the development of asthma [112]. Moreover, an allergic rhinitis model induced by OVA in human nasal epithelial cells showed a decrease in the miR-181a expression and its overexpression reduces the production of TNF-a, IL-1b and IL-6 through the regulation of IL-33 signaling [111]. Therefore, the treatment with Vitamin D3 may regulate the inflammatory response present in asthma through the up-regulation of miR-181a expression.” P. 13-14, L. 569-615.
5. Why you indicate the role of mir in pain? please report only data in asthma.
R= Thank you for your observation. In line to your commentary, in the role of microRNA-146a in asthma and its modulation by vitamin D3 section discussing the role of miR-146a in relation to pain has been removed. The remaining content exclusively addresses its association with asthma.
Round 2
Reviewer 2 Report
Comments and Suggestions for Authors
Dear authors thank you. I have not further comments
Comments on the Quality of English Languagenone